# Comparative Analyses of Grain Quality in Response to High Temperature during the Grain-Filling Stage between $Wx^a$ and $Wx^b$ under *Indica* and *Japonica* Backgrounds

Xiaolei Fan [1,2,3,4], Xiaosong Sun [1,2,3,4], Rui Yang [1,2,3,4], Si Chen [1,2,3,4], Rumeng Li [1,2,3,4], Xinyue Bian [1,2,3,4], Lexiong Xia [1,2,3,4] and Changquan Zhang [1,2,3,4,*]

1   Jiangsu Key Laboratory of Crop Genomics and Molecular Breeding, Yangzhou University, Yangzhou 225009, China
2   Key Laboratory of Plant Functional Genomics of the Ministry of Education, College of Agriculture, Yangzhou University, Yangzhou 225009, China
3   Jiangsu Key Laboratory of Crop Genetics and Physiology, Yangzhou University, Yangzhou 225009, China
4   Jiangsu Co-Innovation Center for Modern Production Technology of Grain Crops, Yangzhou University, Yangzhou 225009, China
*   Correspondence: cqzhang@yzu.edu.cn

**Abstract:** Amylose content controlled by *Wx* determines rice grain quality, which is easily affected by high temperature. $Wx^a$ and $Wx^b$ are the two typical *Wx* alleles in rice, however, their effects on quality formation in response to high temperature under the backgrounds of *indica* rice and *japonica* rice have not been systematically compared. In this study, the near-isogenic lines (NILs) of $Wx^a$ and $Wx^b$ with *japonica* rice 2661 and *indica* rice 3611 backgrounds were treated by high temperature during the grain-filling stages. High temperature accelerated the grain ripening process, decreased the thousand-kernel weight, and increased the chalkiness degree of all rice samples. However, these traits of *Wx* NILs with 3611 background were more susceptible to high temperature than those with 2661 background. Furthermore, high-temperature treatment decreased the amylose contents (AC) and starch viscosities but increased the gelatinization temperature of all the *Wx* NILs. The 3611-$Wx^a$ was atypical $Wx^a$-type rice, whose AC was more sensitive to high temperature. The AC result was consistent with quantitative analysis of GBSSI by Western blot. In addition, the effects of *Wx* genotype and genetic background on rice physicochemical quality (such as the gel consistencies, starch crystallinity, and the morphological structure of starch grains) in response to high temperature were systematically analyzed. These results have important guiding significance for rice-quality improvement under high-temperature climate.

**Keywords:** rice (*Oryza sativa* L.); high temperature; *Wx* allele; amylose; quality

## 1. Introduction

Rice (*Oryza sativa* L.) is one of the main food crops in the world; billions of people worldwide consume polished rice as a staple food, which serves as a rich source of energy because it is composed of up to 90% starch. Hence, the composition and structure of starch are important factors determining rice quality [1]. Rice quality is controlled by both genetic factors and environmental conditions. Environmental temperature is an important factor affecting rice growth and development. In recent years, with the intensification of the greenhouse effect and the acceleration of the industrialization process, the average temperature of the earth continues to rise, and the frequency of extreme high temperatures gradually increases. High temperature has become one of the major disastrous climatic factors affecting rice growth [2]. Whether in the vegetative or reproductive growth period, if the environment temperature exceeds its tolerance limit, it may cause heat damage to rice [3]. Studies have shown that rice yield will decrease by 10% for every 1 °C increase

in temperature [4]. Heat stress will not only make the development of rice flower organs imperfect, but also make rice pollen poorly developed and its vitality decreased. It will also affect the grain filling of rice and increase the number of empty grains. Under high-temperature stress, the head rice rate decreases, chalkiness increases significantly, and cooking and eating quality deteriorate [5]. Starch is the most important component in rice endosperm, existing in the form of granules which are irregular and angular polygons. Starch is composed of amylose and amylopectin [6]. Amylose consists of chain-like glucose molecules with few or no branches which are connected by an $\alpha$-1, 4 glycoside chain [7]. Amylopectin is a highly branched glucose polymer composed of 1,4 glycoside chains in which the distribution of different branches directly affects the structure of starch. The proportion and structure of the two starches determine the quality of rice to the greatest extent [8].

The main reason why high temperature affects rice quality is that temperature during grain filling is closely related to starch composition and the fine structure of rice. Studies show that the influence of temperature on the amylose content of rice is related to the level of amylose content of the variety itself. The amylose content of varieties with low amylose content is positively related to the temperature during filling and setting, while the amylose content of varieties with medium and high amylose content is negatively related to the temperature during filling and setting [9]. The research shows that rice with high amylose content is easy to age after being gelatinized by heat, and will absorb more water during cooking, thus expanding constantly. In the ageing process, the degradation of amylose molecules preferably starts from the shorter amylose chains [10]. It can be seen that one of the important factors affecting the properties of rice starch, such as gelatinization and aging, is the amylose content in starch. Later studies confirmed that starch synthase (E.C.2.4.1.11) is the key enzyme to synthesize amylose, which is encoded by *Wx* and controls the synthesis of amylose in pollen, endosperm, and the embryo sac [11]. Amylose comprises long-chain polydextrose molecules synthesized in endosperm with ADP glucose as substrate under the catalysis of granule-bound starch synthase (GBSSI). Therefore, *Wx* plays a leading role in rice amylose synthesis, and it is a key factor in cooking and eating quality [12]. Rice has evolved into different species in different environments and the variation in quality character is also extremely complex. Amylose content among the existing rice varieties has certain differences. The amylose content of glutinous rice varieties is the lowest (only 0–2%), and the highest amylose content of some rice varieties is more than 30%. These differences are mainly caused by the allelic variation of *Wx*. From cluster analysis of a large number of rice varieties from different sources, it is speculated that *Wx* in rice is mainly divided into two branches, namely, the *Wx*[a] genotype and the *Wx*[b] genotype, which exist in *indica* rice and *japonica* rice, respectively. Subsequently, multiple *Wx* alleles regulating amylose synthesis were identified in different rice varieties, such as *Wx*[in], *Wx*[op], *Wx*[mq], *Wx*[mp], and *Wx*[hp]. The amylose content of rice controlled by different *Wx* alleles was significantly divergent, and the cooking and eating quality of rice was also different [13]. There have been some studies on the effect of high temperature on starch formation during the grain-filling period. *Wx* NILs (near-isogenic lines *Wx*[a], *Wx*[b], *Wx*[op], and *wx*) with *japonica* T65 background were treated at high temperature (32/28 °C) during the filling period. The reduction degree of starch quality of several *Wx* near-isogenic lines affected by high temperature is shown as follows: T65 (*Wx*[op]) > T65 (*Wx*[b]) > T65 (*Wx*[a]) > T65 (*wx*) [14]. It can be seen that different *Wx* alleles have significant differences in response to high temperature for rice quality.

In view of the important decision effect of amylose synthesis on rice quality, *Wx* can be regarded as an important starting point for studying the mechanism of rice-quality formation in response to high temperature. Although some studies have investigated the amylose quality controlled by different *Wx* alleles in response to high temperature, few studies have revealed the effects of different rice genetic backgrounds on amylose synthesis and quality in response to high temperature. The construction and study of *Wx* NILs with different genetic backgrounds are helpful to research the function of *Wx* alleles in response

to high temperature under different genetic backgrounds (Geng and Xian rice). In this study, *Wx* NILs (*Wx*$^a$ and *Wx*$^b$) with *indica*-3611 and *japonica*-2661 rice backgrounds were constructed and used to compare the differences and similarities of amylose synthesis and related quality traits in response to high temperature, hoping to clarify the effect of amylose synthesis and other rice-quality traits in response to high temperature between *indica* and *japonica* background, which provides theoretical basis and experimental basis for the cultivation of high-quality rice materials with high-temperature passivity in the future.

## 2. Materials and Methods

### 2.1. Plant Materials and Heat Treatment

Previously, two *Wx* NILs with 3611 (*Wx*$^a$) and 2661 (*Wx*$^b$) backgrounds were constructed. For 3611 *Wx* NILs, 3611 was used as the recurrent parent and *indica* rice 9311 (*Wx*$^b$) was used as the donor parent in backcrosses. For 2661 *Wx* NILs, 2661 was used as the recurrent parent and *indica* Teqing (*Wx*$^a$) was used as the donor parent in backcrosses (Figure 1A). F$_1$ plants were obtained between donor and recurrent parents and were advanced up to the BC$_9$ generation using marker-assisted selection, and were also resequenced for genetic background analysis (Supplementary Figure S1). Starting from BC$_1$F$_1$, and in each of the following BCF$_1$ generations, approximately 60 plants were genotyped for the selected molecular marker (QRM190) that can distinguish the sequence difference between *Wx*$^a$ and *Wx*$^b$ (Figure 1B). The sequences of the QRM190 primers and the PCR program were as follows: F: 5′-ATTCCTTCAGTTCTTTGTCTATCTCA-3′; R: 5′-TCCTGATGAACAACAGAACAACAC-3′. PCR program: 95 °C for 5 min, 30 cycles of 94 °C for 50 s, 55 °C for 50 s, 72 °C for 40 s, and 72 °C for 10 min. The 2 plants were each planted in 20 L buckets (the soil for cultivation was mixed with urea in the proportion of 1 g urea/3 kg soil) under natural conditions until the rice booting stage (late August). Then, all plants were cultivated in an artificial climate chamber with normal temperature (32 °C/22 °C, light:dark = 14:10; light intensity = 900 μmol.m$^{-2}$.s$^{-1}$; humidity = 50%). At 5 days after flowering, half of the plants were transferred into the artificial climate chamber with high temperature that had the same environmental conditions except for the temperature (38 °C/28 °C). Specific parameter settings of the growth chambers are shown in Supplementary Figure S2.

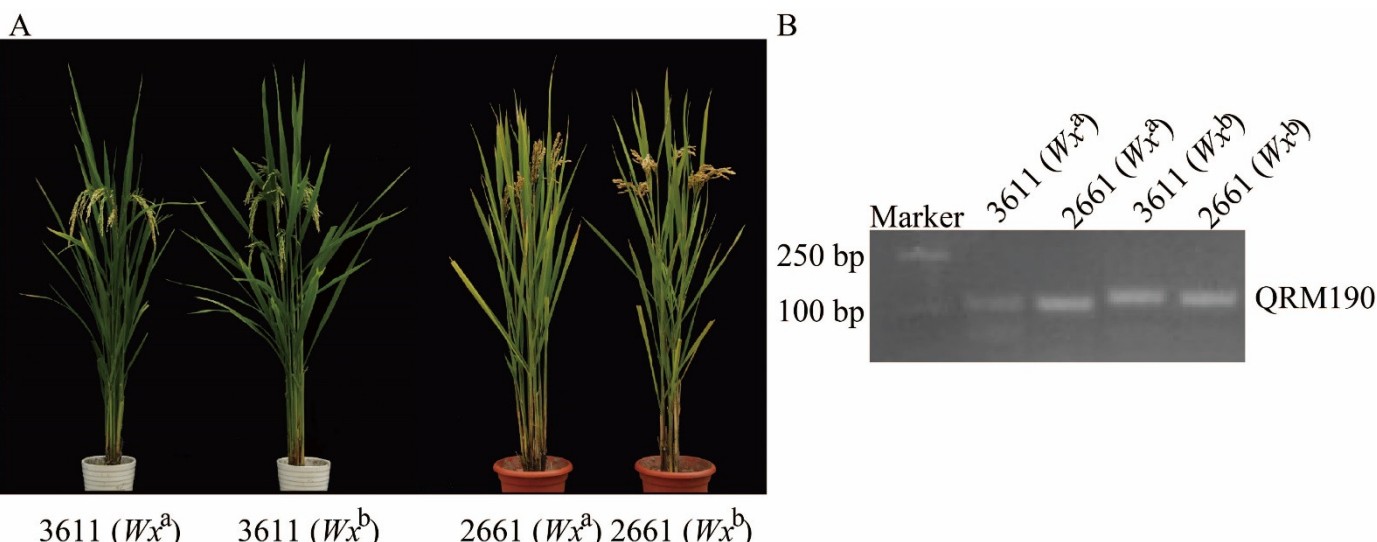

**Figure 1.** Construction of *Wx* NILs with 3611 and 2661 genetic backgrounds. (**A**) The plant phenotype of *Wx* NILs with 3611 and 2661 genetic backgrounds. (**B**) Genotyping of *Wx* alleles in the two *Wx* NILs.

## 2.2. Appearance Characteristics of Rice Grains

The exact flowering time of each rice grain was recorded by using the flowering marking method. Every 5 days (5 d, 10 d, 15 d, 20 d, 25 d after flowering and maturity), the grains of rice materials under high temperature and normal conditions were sampled for later analysis. All sampled seeds had photos taken of them and were observed, and the 1000-grain weight was measured. More than 100 grains of milled mature rice grains were used to measure the chalkiness with a chalk white scanner (MRS-9600TFU2L, MICROTEK, Changzhou, China).

## 2.3. Preparation of Rice Flour and Starch Extraction

According to a neutral protease method with some modifications, polished rice grains were used to isolate rice starches. A 10 g measure of the polished rice grains was soaked in three volumes of Tris-HCL solution (50 mM Tris-HCL, 10 mM $CaCL_2$, pH 7.0). After holding overnight at room temperature, the soaked rice grains were grated using a Waring blender (IKA-T RCT-Basic, Burladingen, Germany) with 1500 rpm for 3 min. Then, 5 mg of protease K (Amresco at VWR, Radnor, PA, USA) was added into the slurry and mixed using a magnetic stirring bar for 24 h at 37 °C. Next, the solution was passed through a 75 μm sifter and the solids retained on the sifter were discarded. After carefully removing the soft top layer of the slurry, the underlying starch layer was centrifuged at $3600\times g$ for 20 min. The sediment was then washed using $dH_2O$ and centrifuged for 15 min after removing the supernatant. The washing step was repeated five times and the starch was placed in a convection oven for drying at 40 °C for 48 h [15].

## 2.4. Observation of Starch Granule Morphology

Before observation, starch samples were pasted onto a circular aluminum specimen stub using double-sided sticky tape and coated with gold. Then, an environmental Scanning Electron Microscope (SEM, Philips XL-30, Philips, Amsterdam, The Netherlands) was used to observe the structure of the native starch granules. The granular size (diameter) was calculated using SEM Image software (Philips).

## 2.5. Starch Microstructure Analyses

Before starch microstructure analysis, purified starch was debranched by isoamylase (EC3.2.1.68, E-ISAMY, Megazyme). Size-exclusion chromatography (SEC) was used to analyze the chain length distributions (CLDs) of amylose. A 2 mg measure of starch was dissolved in DMSO with 0.5% (*w/w*) LiBr at 80 °C for 2 h, which was detected by using Agilent 1100 Series SEC system (Agilent Technologies, Waldbronn, Germany) equipped with GRAM 1000 and GRAM 100 analytical columns (Polymer Standards Service (PSS), Mainz, Germany) set at 80 °C and a differential refractive index (DRI) detector (Wyatt, Santa Barbara, CA, USA). DMSO with 0.5% *w/w* LiBr was used as the eluent and its flow rate was set as 0.6 mL min$^{-1}$. The detailed SEC procedure was described previously [16], and the measurement of apparent amylose content follows our previous study [17].

## 2.6. RVA, DSC, and GC Analyses

A Rapid Visco-Analyser (RVA) (Techmaster, Newport Scientific, Warriewood, Australia) was used to analyze the pasting properties of rice following the previously described methods [18]. A Differential Scanning Calorimeter (DSC; 200 F3, Netzsch Instruments NA LLC, Burlington, MA, USA) was used to detect the gelatinization temperature of starch. A 5 mg measure of starch was weighed accurately and mixed in an aluminum pan with 15 μL $ddH_2O$ water. All samples were placed at 4 °C overnight for balancing after sealing and incubated at room temperature for 1 h before testing. DSC was analyzed according to the previously described methods [19]. Gel consistency (GC) was analyzed as described in a previous study [17], and 100 mg milled rice flour was mixed with 0.2 mL thymol blue indicator and shake by using an oscillator in a long glass tube. After adding 2.0 mL 0.2 mol/L KOH solution, the mixture was boiled in a water bath for 8 min. The mixture of

gelatinized rice flour in glass tube was cooled for 5 min at room temperature and 20 min in ice water, which was placed horizontally for 1 h at room temperature. The length of rice glue reflects the gel consistency of rice.

### 2.7. X-ray Powder Diffraction (XRD)

Starch XRD analysis was carried out using a D8 ADVANCE-type X-ray diffractometer (Bruker, Bremen, Germany), and the relative crystallinity (%) of the starches was measured according to a previously described method [20]. The relative crystallinity = Ic/(Ia + Ic) × 100, where "Ia" is the proportion of the uncrystallized area, and "Ic" is the proportion of the crystallized area in the X-diffraction profile. Before measurements, all the specimens were stored in a desiccator where a saturated solution of NaCl maintained a constant humidity atmosphere (relative humidity = 75%) for 1 week at room temperature.

### 2.8. Statistical Analysis

For characterization of the samples, at least three replicate measurements were performed, unless otherwise specified. Data represent the mean values $\pm$ SD of three independent experiments. Double asterisks denote a highly significant difference obtained using Student's *t*-test ($p < 0.01$). A single asterisk denotes a significant difference obtained using Student's *t*-test ($0.01 < p < 0.05$).

## 3. Results

### 3.1. Dynamic Analysis of Grain Filling

Photographing and weighing rice grains throughout the grain-filling stages under different temperature conditions could intuitively reflect the influence of high temperature on grain grouting. High temperature accelerated the grain-grouting process and resulted in seed precocity in all of the four *Wx* NIL materials, the seed coat of all rice plants lost green, and the photosynthetic process under high temperature ended earlier than those under normal temperature (Figure 2A). In addition, high temperature significantly decreased the grain-filling degrees in both *Wx* NIL materials. Heat treatment significantly decreased the thousand-kernel weight (TKW) of all rice materials. Under the same genetic background, the decrease in the TKW of rice containing *Wx*^a was higher than that of rice containing *Wx*^b. The decrease in the TKW of rice with the same *Wx* genotype (*Wx*^a or *Wx*^b) was higher in 3611 NILs than in 2611 NILs (Figure 2B).

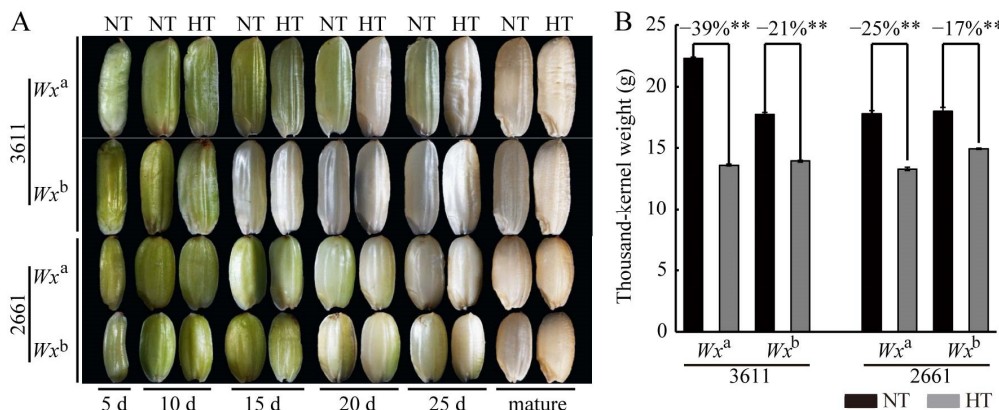

**Figure 2.** Dynamic analysis of grain filling in the *Wx* NILs with 3611 and 2661 genetic backgrounds. (**A**) Grain observation of *Wx* NILs treated by high temperature in different periods by photographing. "NT" stands for normal condition and "HT" stands for high-temperature condition. (**B**) Thousand-kernel weight of four materials treated by different temperatures (grains not fully ripe were also collected to measure weight). Double asterisks denote a highly significant difference using Student's *t*-test ($p < 0.01$). All significant-difference analyses were performed between the same samples at different temperatures.

### 3.2. Calculation of Grain Chalkiness Ratio

Under the same genetic background, the transparency of grain from rice containing $Wx^b$ is higher than that from rice containing $Wx^a$ at normal temperature. Furthermore, the transparency of grain from rice containing the same $Wx$ allele is higher in 3611 than that in 2661. Heat treatment resulted in higher chalkiness degrees and yellow aleurone layer in all four rice materials (Figure 3A). Under the same genetic background, the ascensional range of chalkiness degree by high temperature is higher in $Wx$ NILs containing $Wx^b$ than those containing $Wx^a$. Moreover, the ascensional range of chalkiness degree by high temperature is higher in 3611 NILs than that in 2661 NILs which contain the same $Wx$ allele (Figure 3B).

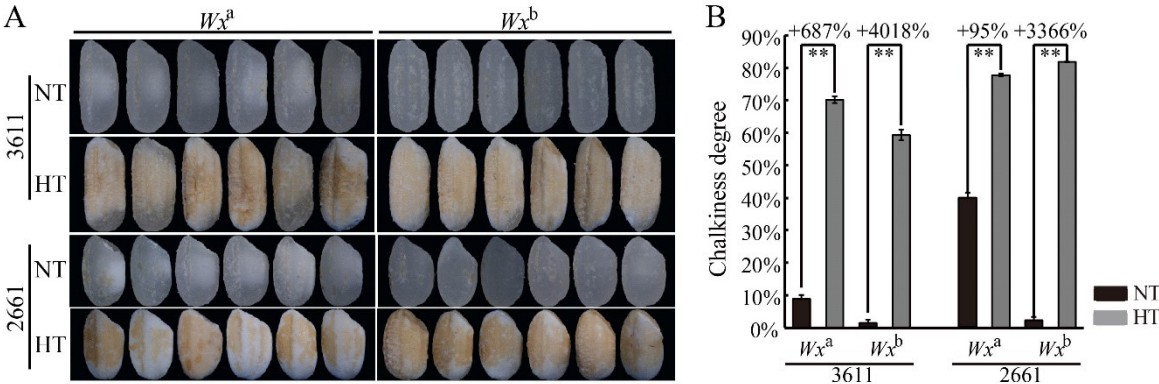

**Figure 3.** The influence of high temperature on the grain-filling degree of different $Wx$ NILs. (**A**) Observation of mature grains from different $Wx$ NILs treated by high temperature by photographing. (**B**) Chalkiness ratio of mature seeds from different $Wx$ NILs treated by high temperature. "NT" stands for normal condition and "HT" stands for high-temperature condition. Double asterisks denote a highly significant difference using Student's *t*-test ($p < 0.01$). All significant-difference analyses were performed between the same samples at different temperatures.

### 3.3. Observation of Starch Granule Morphology

Several studies have proven that the size and shape of starch granules could affect the starch rheological properties. SEM was used to observe the starch grain morphology (Figure 4A), and starch granulometric distributions were further investigated (Figure 4B). The results indicated that the starch granulometric distributions in response to high temperature were similar between $Wx$ NILs under the same genetic background, but the change trends were different between $Wx$ NILs under 3611 and 2661 backgrounds. Under the 3611 background, high temperature resulted in narrower curves of starch granulometric distributions with higher peaks. The starch granulometric distribution curve peak of 3611-$Wx^a$ moved forward under high-temperature condition, whereas that of 3611-$Wx^b$ did not move. This indicated that high temperature increased the content of starch with medium–long chains but decreased the diameters of most abundant starch grains in 3611-$Wx^a$. Under a 2661 rice background, the starch granulometric distribution curve peak of $Wx$ NILs moved backwards under high-temperature conditions, and the size of the backward move of 2661-$Wx^b$ was more significant. This indicated that high temperature increased starch grain size in 2661-$Wx^b$.

### 3.4. Determination of Starch Fine Structure

To investigate the starch fine structure in response to high temperature, SEC technology was used to identify the chain-length distribution of starch from the four rice materials planted under different temperature conditions. The four rice materials contain amylose and amylopectin; therefore, all the SEC curves present three peaks, including P1, P2, and P3. Under normal temperature, the starch chain-length distributions from NILs with the same $Wx$ allele ($Wx^a$ or $Wx^b$) were similar between 3611 and 2661. Amylose contents of $Wx^a$ NILs were higher than those of $Wx^b$ NILs, while the propor-

tions of amylopectin were lower in $Wx^a$ NILs than those in $Wx^b$ NILs. High temperature led to different decrease extents of amylose content of all $Wx$ NILs except 2661-$Wx^a$ (Figure 5A–D and Supplementary Table S1). The results of SEC were basically consistent with those of apparent amylose content and GBSSI quantitative analysis (Figure 5E and Supplementary Figure S3). Compared with the normal-temperature condition, the GBSSI protein content in the mature seeds of four $Wx$ NILs decreased by different extents under high temperature. The GBSSI protein content in 2661-$Wx^a$ seeds was the least affected by high temperature, while that in 3611-$Wx^a$ seeds was the most affected by high temperature. The contents of amylopectin with short chains increased remarkably in all four $Wx$ NILs under high temperature. By contrast, the proportions of amylopectin with medium length chains only increased in $Wx$ NILs with 3611 background, while the content of amylopectin with medium-length chains significantly decreased in 2611-$Wx^a$ by high temperature, and that of 2611-$Wx^b$ was not affected.

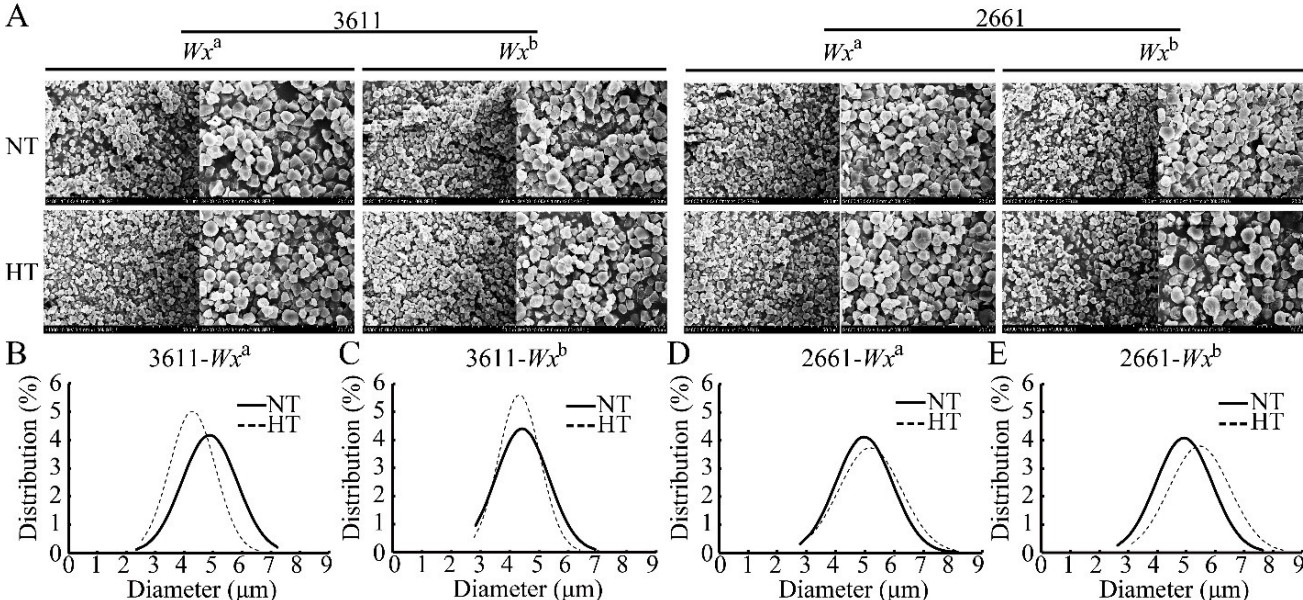

**Figure 4.** Observation and analysis on the starch granules of different $Wx$ NILs under different temperature conditions. (**A**) SEM observation on the starch granules from four rice materials planted under different temperature conditions. (**B–E**) The influence of high temperature on the starch granule size distributions different $Wx$ NILs. "NT" stands for normal condition and "HT" stands for high-temperature condition.

*3.5. Identification of Starch Viscosity*

The RVA profile is one of the important factors controlling rice cooking and eating quality evaluation. RVA properties include peak paste viscosity (PKV), hot paste viscosity (HPV), cool paste viscosity (CPV), breakdown viscosity (BDV), setback viscosity (SBV), peaktime (PeT), and pasting temperature (PaT). Under normal temperature, PKV, CPV, HPV, BDV, and SBV were higher in the NIL of $Wx^b$ type than that of $Wx^a$ type with 2661 background. Most viscosity properties of 3611 NILs were similar with 2661 NILs except HPV and SBV, which were lower in the NIL of $Wx^b$ type than that of $Wx^a$ type (Table 1). The RVA profile curves of rice powder from all rice materials planted under high-temperature conditions were lower than those planted under normal conditions, which indicated that the PKV, HPV, and CPV of all rice materials decreased significantly under high-temperature conditions. However, there were significant differences in the degree of viscosity response to high temperature among the four rice starches (Supplementary Figure S4). Both in 3611 and 2661 backgrounds, HPV and CPV of NILs containing $Wx^b$ decreased by about 40% under high temperature. The decrease extent of 2661-$Wx^a$ PKV was about twice that of 3611-$Wx^a$. By contrast, the decrease extent of 3611-$Wx^a$ HPV was about four times that of

2661-$Wx^a$. The decrease extent of 3611-$Wx^a$ HPV was about 20 times of that of 2661-$Wx^a$. BDV of 3611 $Wx$ NILs and 2661-$Wx^a$ decreased, while that of 2661-$Wx^b$ increased under high temperature. SBV of NILs containing $Wx^a$ increased, while that of NILs containing $Wx^b$ decreased under high temperature (Table 1).

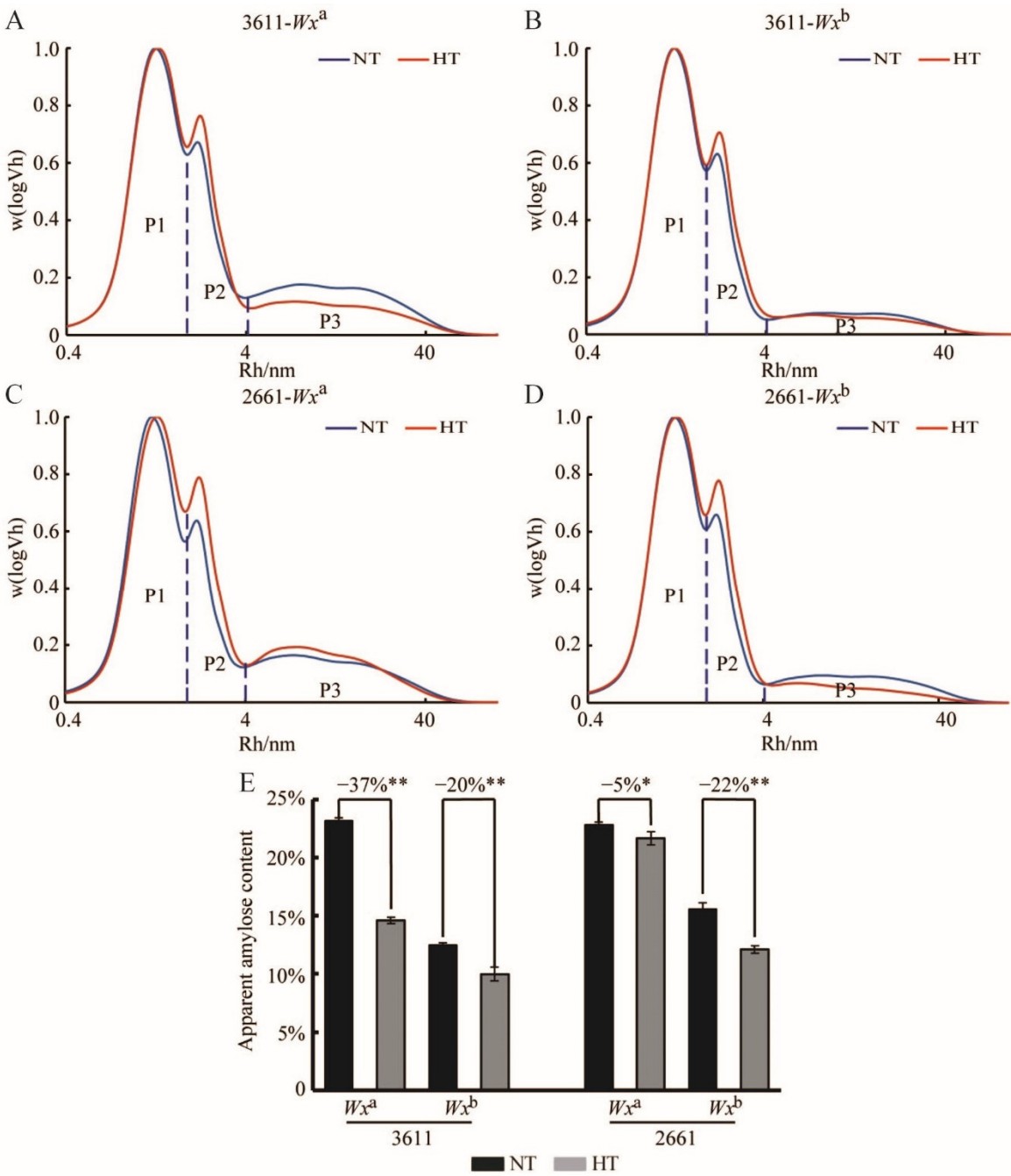

**Figure 5.** Starch SEC analysis of $Wx$ NILs with 3611 (**A**,**B**) and 2661 (**C**,**D**) genetic backgrounds under different temperature conditions. P1, P2, and P3 represent the peaks of grade 2 amylopectin, grade 1 amylopectin, and amylose, respectively. P1 is composed of A chains and short B chains of external starch, and P2 is composed of long B chains. (**E**) Apparent amylose content analysis of $Wx$ NILs with 3611 and 2661 backgrounds. Double asterisks denote a highly significant difference using Student's *t*-test ($p < 0.01$). A single asterisk denotes a significant difference using Student's *t*-test ($0.01 < p < 0.05$).

**Table 1.** RVA analysis of different *Wx* NILs planted under different temperature conditions.

| Sample | Treat | PKV (Cp) | HPV (Cp) | BDV (Cp) | CPV (Cp) | SBV (Cp) | PeT (min) | PaT (°C) |
|---|---|---|---|---|---|---|---|---|
| 3611-*Wx*<sup>a</sup> | NT | 2948 ± 2.83 d | 2733 ± 1.41 b | 217 ± 2.12 e | 3266 ± 5.66 c | 319 ± 2.12 d | 6.90 ± 0.14 b | 74.67 ± 0.40 f |
| | HT | 1590 ± 3.54 h | 1470 ± 6.36 g | 124 ± 2.83 g | 2002 ± 2.83 h | 415 ± 2.12 b | 7.30 ± 0.14 a | 84.70 ± 0.14 b |
| 3611-*Wx*<sup>b</sup> | NT | 3750 ± 2.83 a | 2726 ± 3.54 b | 1021 ± 4.24 b | 3785 ± 4.24 b | 35 ± 1.41 e | 6.55 ± 0.07 c | 76.60 ± 0.00 d |
| | HT | 2557 ± 3.54 f | 1597 ± 2.83 f | 961 ± 2.12 c | 2208 ± 2.83 g | −345 ± 4.24 f | 6.15 ± 0.07 d | 84.40 ± 0.07 b |
| 2661-*Wx*<sup>a</sup> | NT | 2763 ± 2.83 e | 2585 ± 3.54 c | 180 ± 2.12 f | 3140 ± 2.83 d | 375 ± 2.12 c | 7.15 ± 0.07 a | 74.50 ± 0.71 f |
| | HT | 2378 ± 0.00 g | 2331 ± 0.00 d | 47 ± 0.00 h | 3079 ± 0.00 e | 701 ± 0.00 a | 7.20 ±0.14 a | 86.17 ± 0.03 a |
| 2661-*Wx*<sup>b</sup> | NT | 3523 ± 4.95 b | 2867 ± 3.54 a | 653 ± 2.83 d | 3930 ± 2.83 a | 412 ± 3.54 b | 6.74 ± 0.01 b | 75.48 ± 0.10 e |
| | HT | 3470 ± 7.07 c | 1690 ± 2.12 e | 1781 ± 4.95 a | 2218 ± 0.00 f | −1254 ± 4.24 g | 5.43 ± 0.06 e | 83.71 ± 0.06 c |

Note: "NT" stands for normal condition and "HT" stands for high-temperature condition. RVA properties include peak paste viscosity (PKV), hot paste viscosity (HPV), cool paste viscosity (CPV), breakdown viscosity (BDV), setback viscosity (SBV), peaktime (PeT), and pasting temperature (PaT). The data presented in the table are repeated through three independent experiments, and values with different letters (a–g) in the same column are significantly different at $p < 0.05$, calculated by one-way ANOVA.

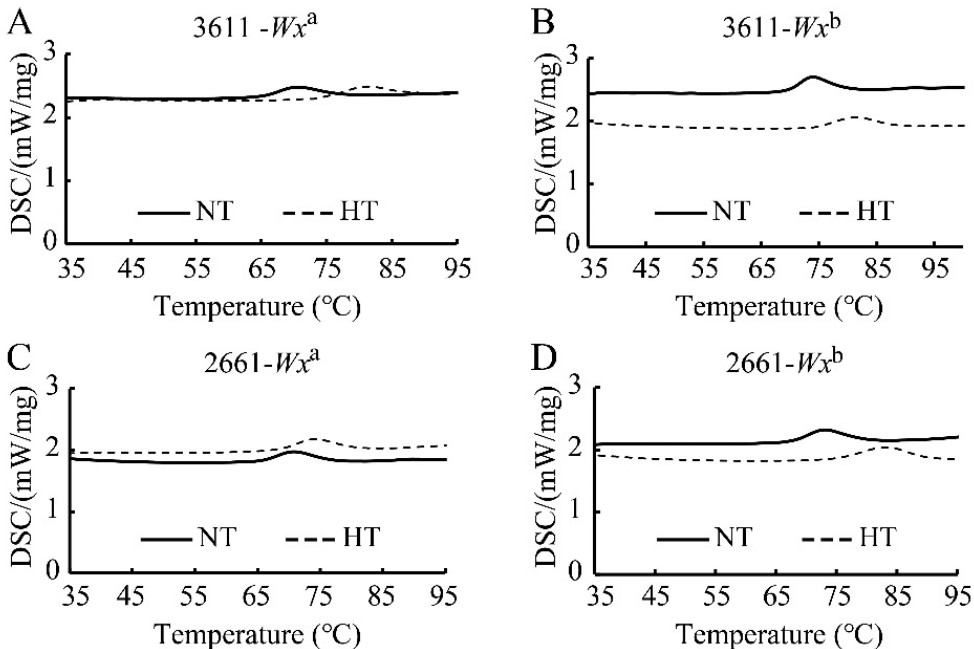

**Figure 6.** The influence of high temperature on the gelatinization features of *Wx* NILs with 3611 (**A**,**B**) and 2661 (**C**,**D**) genetic backgrounds. "NT" stands for normal condition and "HT" stands for high-temperature condition.

*3.6. Measurement of Gelatinization Temperature*

To quantitatively analyze the thermal properties of rice flour in aqueous solution, DSC was used to characterize the rice grains grown under different temperature conditions. The DSC curve indicated that there was a significant difference in the gelatinization properties of the rice grains grown under different temperature conditions. The thermomechanical curves of all four materials presented an obvious endothermic peak of starch melting (Figure 6). All parameters of thermal properties including the Tp, To, Tc, and $\triangle H_{gel}$ are listed in Supplementary Table S2. Almost all DSC parameters of these rice materials increased under high-temperature conditions. Under the 3611 genetic background, the influence of high temperature on these DSC parameters was more significant in rice containing *Wx*<sup>a</sup> than that rice containing *Wx*<sup>b</sup>. Under the 2661 genetic background, the influence of high temperature on these DSC parameters was more significant in rice containing *Wx*<sup>b</sup> than in rice containing *Wx*<sup>a</sup>. In addition, the increase in DSC parameters caused by heat treatment was larger in 3611 rice containing *Wx*<sup>a</sup> than in 2661 rice containing *Wx*<sup>a</sup>. The increase in DSC parameters caused by heat treatment was higher in 2661 rice containing *Wx*<sup>b</sup> than in 3611 rice containing *Wx*<sup>b</sup>.

### 3.7. Gel-Consistency Determination

Gel consistency (GC) is an important index to evaluate rice cooking and quality. The GC value is closely related to the degree of hardness of rice. Rice with low amylose content has low viscosity and the cooked rice is soft, and many *japonica* rice are of this type. *Indica* rice has relative high amylose content; therefore, its GC is generally high, and the cooked rice is relatively hard. Under both genetic backgrounds, the GC of rice containing $Wx^a$ was not affected by high temperature, while that of rice containing $Wx^b$ decreased significantly under heat treatment. The GC decrease extent of 2661-$Wx^b$ was larger than that of 3611-$Wx^b$ (Figure 7).

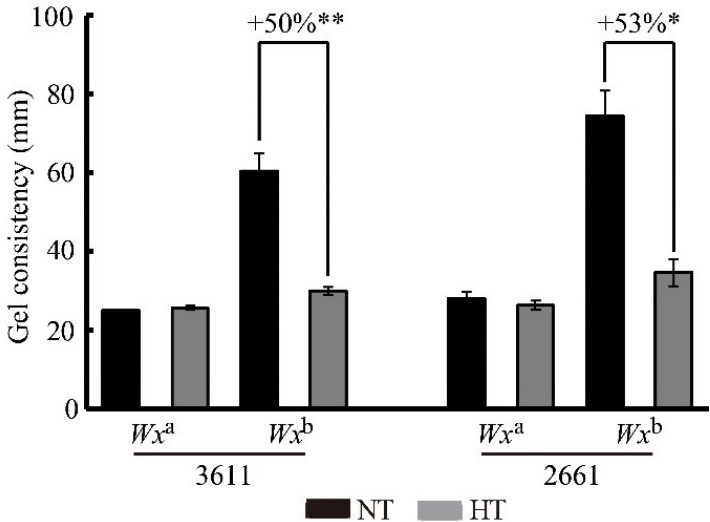

**Figure 7.** The influence of high temperature on the gel consistency of the mature seed flour from different *Wx* NILs. "NT" stands for normal condition and "HT" stands for high-temperature condition. Double asterisks denote a highly significant difference using Student's *t*-test ($p < 0.01$). A single asterisk denotes a significant difference using Student's *t*-test ($0.01 < p < 0.05$). All significant-difference analyses were performed between the same samples at different temperatures.

### 3.8. Starch Crystallinity Analysis

The XRD diffraction pattern of native starches from the four rice materials planted under different temperature conditions is shown in Figure 8. Starches from all four rice materials planted under different temperature conditions showed the typical A pattern, with main peaks at diffraction angles 2θ of 15°, 17°, 18°, and 23°, as in normal cereal starches. Under normal temperature, the starch relative crystallinity of NILs containing $Wx^b$ was significantly higher than that of NILs containing $Wx^a$ no matter in 3611 or in 2661. The starch relative crystallinity of 3611-$Wx^b$ was similar with 2661-$Wx^b$, while the starch relative crystallinity of 2661-$Wx^a$ was higher than that of 3611-$Wx^a$. High temperature increased the starch relative crystallinity of all four rice materials (Supplementary Table S3). By comparation, the increase extent of starch relative crystallinity by high temperature was highest in 3611-$Wx^a$, while that of other *Wx* NILs was much smaller.

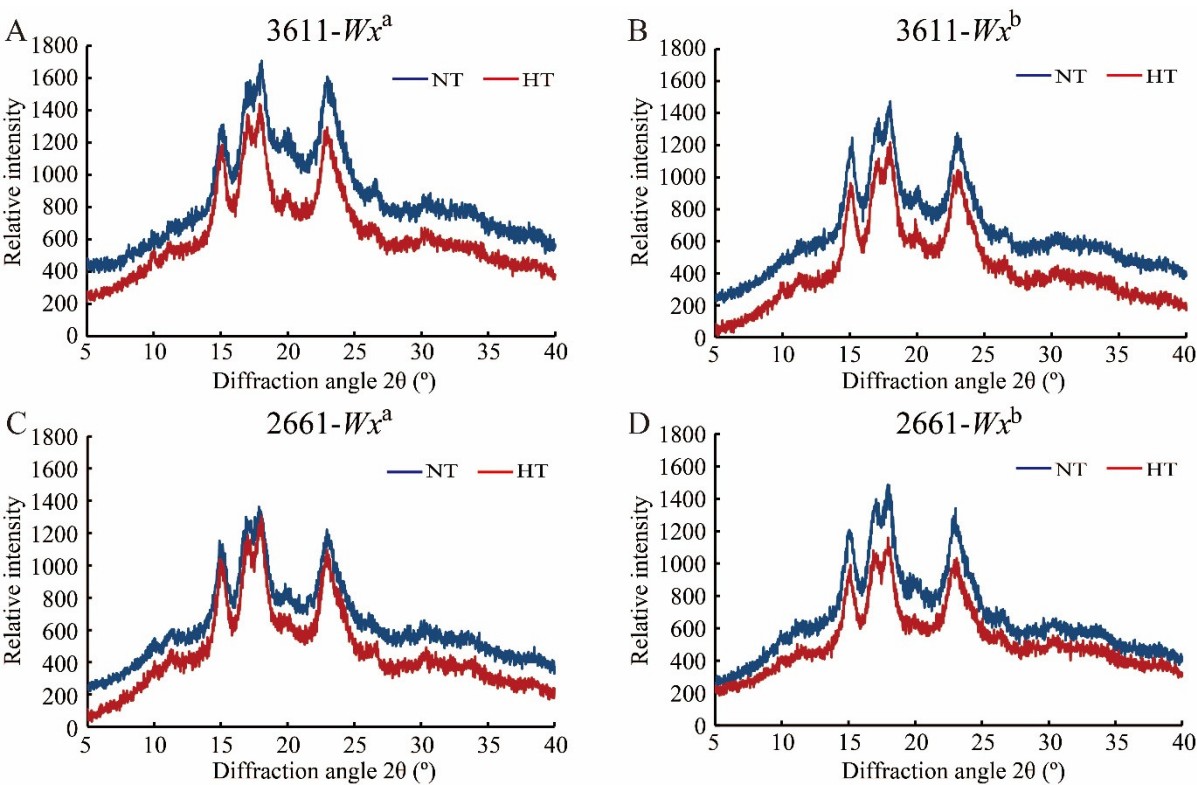

**Figure 8.** XRD spectra of different *Wx* NILs starches with 3611 (**A**,**B**) and 2661 (**C**,**D**) genetic backgrounds under different temperature conditions. "NT" stands for normal condition and "HT" stands for high-temperature condition.

## 4. Discussion

Rice quality mainly forms during the grouting period; this process is sensitive to environment conditions. High temperature during the grain-filling stage is the key factor causing a decrease in rice production and grain quality [21]. These results are consistent with our research. High temperature accelerated the process of rice grain filling, increased the chalkiness ratio, and decreased thousand-kernel weight in all four rice materials. By contrast, thousand-kernel weights of NILs with 3611 background containing the same *Wx* allele (*Wx*$^a$ or *Wx*$^b$) were more sensitive to high temperature. Among them, the thousand-kernel weight of 3611-*Wx*$^a$ was most sensitive to high temperature. As grains not fully ripened were also included for weight measurement, we concluded that the brown rice of 3611-*Wx*$^a$ underwent intense growth and stopped growing during enlargement under long high-temperature treatment. The gel consistency of Xian and Geng rice with *Wx*$^a$ was unaffected by high temperature, while those with *Wx*$^b$ had a decreased GC and this decrease in Xian rice was more remarkable. High temperature decreased the BDV and increased the SBV in both Xian and Geng rice with *Wx*$^a$. While the BDV and SBV in response to high temperature were totally different in Xian and Geng with *Wx*$^b$. In general, high BDV and low SBV contribute to high eating quality of rice. Hence, an appropriate high temperature probably decreased the eating quality of *Wx*$^a$-type rice but increased the eating quality of *Wx*$^b$-type rice. All parameters of thermal properties including Tp, To, Tc, and $\triangle$H$_{gel}$ were significantly increased by high temperature, making rice starch hard to gelatinize. This is considered as the main reason for the decrease in rice cooking quality under high-temperature treatment.

Starch features are a key factor determining rice grain quality and its response to high temperature. Previous studies showed that high temperature accelerated starch granule development and changed the shape and arrangement of starch cells in early *indica* rice grains. Under natural conditions, the starch grains in the grain were closely bound, and

many starch grains were stuck together and combined into clusters, while under the high-temperature treatment, the starch grains were loosely bound, and most of the starch grains in the grain were in the form of single starch grains, and the refractive index and endosperm transparency decreased, leading to chalky formation [22]. In this study, the variation trend of starch grain size in rice seeds of *Wx* NILs in responding to high temperature was completely opposite between the *indica* and *japonica* background. Starch grain size in the *Wx* NILs with 3611 background was significantly decreased, while that in 2661 background was significantly increased under high temperature. In addition, the changes in grain size in response to high temperature were also significantly different between NILs containing $Wx^a$ and $Wx^b$ with the same genetic background. The starch particle size of $Wx^a$ and $Wx^b$ NILs under the background of 2661 had the same variation trend (the particle size became larger) under high temperature, while those under the 3611 background had a slightly different variation trend between NILs containing $Wx^a$ and $Wx^b$. The starch particle size of 3611-$Wx^a$ decreased overall. However, the grain size of starch grains with the largest proportion in 3611-$Wx^b$ did not change, but their proportion became higher. Some research has proved that high-temperature treatment during grain-filling period does not affect the crystal type of rice starch grain, but changes the angle between glucose bases on the $α$-l, 6 glycosidic bond of the double helix formed by amylopectin in starch crystalline regions, which implies that high temperature increases relative crystallinity of starch by affecting crystallite structure of starch. This study further confirmed the above conclusion; however, there were significant differences in the response of starch crystallinity to high temperature between the two *Wx* alleles in different genetic backgrounds. The increase extent of starch crystallinity under high temperature was higher in 3611-$Wx^a$ than that of 3611-$Wx^b$, while the increase extent of starch crystallinity under high temperature was higher in 2661-$Wx^b$ than that of 2661-$Wx^a$. Previous studies indicated that there is a strong negative correlation between relative crystallinity and amylose content [23], which is consistent with the results of amylose content in this study. The decline degree of amylose content under high temperature was higher in 3611-$Wx^a$ than that of 3611-$Wx^b$, while the decline degree of amylose content under high temperature was lower in 2661-$Wx^a$ than that of 2661-$Wx^b$.

Amylose and amylopectin are two forms of starch. Their chain-length distribution can influence the structure of starch grain, which affects the cooking and eating quality of rice. There is a very regular balance between amylose and amylopectin synthesis which can easily be broken by high temperature. Disordered starch composition and arrangement directly lead to poor rice quality [24]. Much research indicates that amylose content of $Wx^a$ type is insensitive to high temperature, and hence 2611-$Wx^a$ belongs to typical $Wx^a$ rice while 3611-$Wx^a$ is atypical. Amylose is synthesized by GBSSI encoded by the *Wx* gene, and the pattern of *Wx* alleles' expression in responding to high temperature determines the amylose synthesis under high temperature. The expression of GBSSI encoded by 2611-$Wx^a$ was much more stable than that encoded by 3611-$Wx^a$. It has been reported that the effects of high temperature on amylose content are variety-dependent. Amylose content in rice endosperm is not only decided by GBSSI, but also influenced by the activities of other starch synthesis-related enzymes, such as SDBE, SBE, ADPG-Ppase, and SuSy, which are different among various varieties [9]. In addition to amylose content, other physicochemical qualities of 3611-$Wx^a$ were easily affected by high temperature. Hence, the special regulation mechanism of $Wx^a$ expression responding to high temperature in 3611 deserves further study. Previous studies indicated that the influences of high temperature on amylose content controlled by $Wx^b$ were different between *indica* rice 9311 and *japonica* rice NPB. High temperature remarkably decreased the amylose content in NPB ($Wx^b$), while the decline extent of amylose content in 9311 ($Wx^b$) was much smaller than that of NPB ($Wx^b$) [25,26]. In our study, the decline extent of amylose content in 3611-$Wx^b$ was about half of that in 2661-$Wx^b$, which is similar to the results of the above studies. The underlying molecular mechanism such as the regulation of *Wx* expression remains to be further studied. Considering that 3611 and 2661 are important parental materials in rice

breeding, this study has important guiding significance for breeding design of rice with high quality under global warming climate conditions, such as breeding parent selection in different regions.

**Supplementary Materials:** The following supporting information can be downloaded at: https://www.mdpi.com/article/10.3390/agronomy13010017/s1, Figure S1: Genomic resequencing analysis of *Wx* NILs with different backgrounds; Figure S2: Artificial climate chamber condition setting for temperature treatment; Figure S3: The influence of high temperature on the expression level of GBSSI protein in the mature seeds from different *Wx* NILs; Figure S4: The influence of high temperature on the RVA features of *Wx* NILs with 3611 and 2661genetic backgrounds; Table S1: Starch SEC analysis of different *Wx* NILs planted under different temperature conditions; Table S2: DSC analysis of different *Wx* NILs planted under different temperature conditions; Table S3: Relative crystallinity of different *Wx* NILs planted under different temperature conditions.

**Author Contributions:** Investigation, X.S., R.Y.; S.C., R.L., X.B. and L.X., data curation, X.F., writing—review & editing, C.Z.; supervision and project administration, X.F., funding acquisition, X.F.; Formal Analysis, X.S. and X.F.; writing—original draft, X.F. All authors have read and agreed to the published version of the manuscript.

**Funding:** This study was funded by the National Science Foundation of China: 32072032 and U19A2032; Projects from the government of Jiangsu Province: JBGS (2021) 001, CX(21)3010, BZ2021017, BK20200045, KJRH202208 and KYCX21_3246.

**Informed Consent Statement:** Not applicable.

**Data Availability Statement:** Not applicable.

**Acknowledgments:** We sincerely thank Jindong Wang for their help in data analysis.

**Conflicts of Interest:** The authors declare no conflict of interest.

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
