# Peer review of "Comparative Analyses of Grain Quality in Response to High Temperature during the Grain-Filling Stage between Wxa and Wxb under Indica and Japonica Backgrounds"

_agronomy, doi:10.3390/agronomy13010017_

Round 1
Reviewer 1 Report
The Manuscript compared the Wxa and Wxb alleles effects on the rice grain quality under high temperature treatment. The two alleles were integrated into two rice breeding varieties by constructing near-isogenic lines (NILs), the Japonica 2611 Wxa/Wxb and indica 3661 Wxa/Wxb. The authors chose to treat the NILs under 38℃ at rice booting stage and 28℃ as the control temperature with the other same environment chamber growing. The research analysed the grain quality by using a series of food chemistry technologies. The results indicated the high temperature treatment accelerate the grain ripening, increase the chalky, decrease the amylose content and the starch viscosity, increase the gelatinization temperature etc. And the two Wx alleles in two subspecies showed different effects on the grain cooking and eating quality. The results indicate under high temperature treatment, the Wxa alleles is more affected by high temperature than Wxb. And NILs at indica background were more susceptible to high temperature then NILs at Japonica background. The results will be as important reference to be directly used in rice breeding by marker selection in future.
Some comments as follows:
1. The results are good with replicates. But the authors just present the results at each part with less focus on the results summary and raised up the highlight point, which is important to toward quality impact.
2. Some parts of the results writing are too complicated to understand, simplify the sentence and a bit polish is needed.
3. Gel consistency results indicated the NIL-Wxa lines have no affect to the GC values, but the amylose content already decreased. Dose this affect by the variety background, Wxa expression or other reasons?
Minor concerns as follows:
1. why chose the high temperature 38 ℃ as the high temperature treatment?
2. Line 135: Starch extraction has no reference paper
3. Line 224: what the “yellow aleurone layer” means in here? did the polish process still not get rid of aleurone layer, so the colour is yellow but not white?
4. Line 267-268: “The influences of high temperature on starch fine structure were similar between 3611-Wxa and 2661-Wxb, as well as between 3611-Wxb and 2661-Wxa (Supplementary Table 1).” Very hard to understand the sentence.
5. Hard to understand 3.4—please simplify the words and summarize the results.
6. Line 310: This part should 3.6 but instead of 3.5, following part orders should be changed.
7. Line 331: Red alphabet in “The”
8. Line 382-383: What the sentence meaning “It has been reported that the influence of temperature on rice amylose content depends on the level of rice amylose content. Amylose content is negatively related to the temperature of the grain-filling stages in rice varieties with a low amylose content, while amylose content is positively related to temperature of grain-filling stages in rice varieties with a high amylose content.”
9. Line 398: what the “grouting degree” means? What this relative with the results in this manuscript?
10. Line 412: what means of “starch development” in here.
11. Line 450: please list the “Other starch properties”
12. Lin494: In reference, there missed or misplaced the journal name, volume, pages et al. And the font and format were different in the reference list. Author list are not the same format. Please read Agronomy Journal reference format requirement. Such as no Journal name of the 4th ,6th ,11st, 14th, 15th, 16th, 17th, 18th, 23rd 25th ,26th reference paper etc.
Author Response
Dear reviewer,
Many thanks for your constructive suggestions of our research, which is very necessary to improve the scientific rigor and readability of the manuscript. We have made serious revisions to our manuscript. Responses to reviewer are as follows:
- The results are good with replicates. But the authors just present the results at each part with less focus on the results summary and raised up the highlight point, which is important to toward quality impact.
Response: We have summarized and condensed some of the results, which were marked with yellow color in the manuscript.
- Some parts of the results writing are too complicated to understand, simplify the sentence and a bit polish is needed.
Response: We have rewritten some contents of the result.
- Gel consistency results indicated the NIL-Wxa lines have no affect to the GC values, but the amylose content already decreased. Dose this affect by the variety background, Wxaexpression or other reasons?
Response: In our previous studies, amylose content and gel consistency (rice glue length) of most rice varieties were negatively correlated at normal temperature. But there were some exceptions, for example, the two parameters are positively correlated with each other in the rice varieties with Wxlv. We speculate that this may be caused by gelatinization features of starch grains, which is influenced by the expression of Wx, interaction among starch synthesis related enzymes, as well as their responses to high temperature. These features are different among rice varieties.
Minor concerns as follows:
- why chose the high temperature 38 ℃ as the high temperature treatment?
Response: In this study, the dynamic temperature conditions similar to the summer high temperature in Yangzhou (>35℃) were used to treat rice plants during grain-filling stage, which would be more instructive for rice breeding.
- Line 135: Starch extraction has no reference paper
Response: A reference paper has been cited in the part of starch extraction description (line 153).
- Line 224: what the “yellow aleurone layer” means in here? did the polish process still not get rid of aleurone layer, so the color is yellow but not white?
Response: The long-time of high temperature treatment during the whole filling period in this study resulted in the thickening of aleurone layer in all rice grains. In order to comprehensively present the effect of high temperature on rice grain, grains with partially unpolished grains were exhibited our manuscript. Completely-polished grains (get rid of the yellow aleurone layer) were used for physical and chemical quality analysis but not for photography, because deep polishing resulted in significantly smaller broken grains, which are not suitable for photography. If it may cause misunderstanding of readers, we can consider deleting this part of data.
- Line 267-268: “The influences of high temperature on starch fine structure were similar between 3611-Wxa and 2661-Wxb, as well as between 3611-Wxb and 2661-Wxa (Supplementary Table 1).” Very hard to understand the sentence.
Response: We have deleted this sentence that is hard to understand.
- Hard to understand 3.4—please simplify the words and summarize the results.
Response: We have simplified the words and summarized the results of the part of 3.4 (line268-282).
- Line 310: This part should 3.6 but instead of 3.5, following part orders should be changed.
Response: We have corrected the part order here (line 315).
- Line 331: Red alphabet in “The”
Response: We have corrected the color of the font (line 336).
- Line 382-383: What the sentence meaning “It has been reported that the influence of temperature on rice amylose content depends on the level of rice amylose content. Amylose content is negatively related to the temperature of the grain-filling stages in rice varieties with a low amylose content, while amylose content is positively related to temperature of grain-filling stages in rice varieties with a high amylose content.”
Response: We have rewritten this sentence as follows: It has been reported that the effects of high temperature on amylose content was variety-dependent. Amylose content in rice endosperm is not only decided by GBSSâ… , but also influenced by the activities of other starch synthesis related enzymes, such as SDBE, SBE, ADPG-Ppase and SuSy, which are different among various varieties (line 432-436).
- Line 398: what the “grouting degree” means? What this relative with the results in this manuscript?
Response: We have replaced the “grouting degree” by “thousand kernels weight”, which is relative with figure 2B (line 377).
- Line 412: what means of “starch development” in here.
Response: We have replaced the “starch development” by “starch granule development” (line 390-391).
- Line 450: please list the “Other starch properties”
Response: We have deleted the “Other starch properties” (line 436).
- Lin494: In reference, there missed or misplaced the journal name, volume, pages et al. And the font and format were different in the reference list. Author list are not the same format. Please read Agronomy Journal reference format requirement. Such as no Journal name of the 4th, 6th, 11st, 14th, 15th, 16th, 17th, 18th, 23rd 25th ,26th reference paper etc.
Response: We have reformatted the references according to the journal format requirement (line 481-541).

Reviewer 2 Report
This study investigated the difference in responses to high temperature between Wxa and Wxb under two genetic backgrounds. I acknowledge that this report has a certain degree of originality. However, it is doubtful whether the experiments were conducted appropriately in the following two points.
1. Are the NILs used in the experiments truly Near Isogenic Lines?
The authors claim that they constructed two NILs using one molecular marker (L110-113). Alleles of the Wx gene can be identified with only one marker, but it is impossible to distinguish how far the donor parent's chromosomal region extends. If the authors used only one marker for the selection of NILs, there is no guarantee that the NIL will be produced even if nine rounds of BC are performed. Information needs to be given as to how much of the donor parent's chromosomal region these NILs have.
2. Were the rice grains properly polished?
In the photograph of Fig. 3A, all grains in HT condition have brown areas. These grains don't seem to be polished enough. Machine for measuring the chalkiness degree of grains generally identifies white immaturity areas based on the light transmitted through the grains. Accurate measurement is impossible when a large amount of rice bran remains in samples. The data on the Chalkiness degree in the HT condition are unreliable.
The important problems of this manuscript are pointed out below.
L104-122
Not enough information on growing conditions for plant materials. Write down the sowing date (or the transplanting date), heading date for each line, amount of applied fertilizer, bucket size, and number of plants grown per bucket. The minimum temperature (or average temperature) in the growth chamber should be indicated in addition to the maximum temperature.
L198-200 “Under normal temperature, ... from 2661-Wxb.”
The only data available were photographs (Fig. 2A), and I was unable to read the results claimed by the author from these photographs. The authors should present the results objectively by quantifying the length, width, weight, etc. of the growing grains.
Fig. 2B
While the 1,000-grain weight of the 3611 background Wxa in NT is about 22 g, the weight in HT appears to be about only 14 g. In general, normally ripened rice grains don’t produce such a large difference in 1,000-grain weight. It can be inferred that the 1,000-grain weight was measured including the rice that had stopped ripening. Thousand grain weight should be measured by fully ripened rice. Sterile grains should be removed and remeasured.
L217-220 “The chalky area of ..., which lead to chalky production.”
This is not the content that should be noted in the results.
Fig. 6
Data such as BD and PK shown in Supplementary Table 2 are important for the analysis of starch properties using RVA. This figure should be moved to Supplementary data and Supplementary Table 2 should be included in the text. However, the experimental data in Supplementary Table 2 do not appear to have been measured multiple times. Perform replicate experiments and show the mean and the results of the significance test.
L362-393
This paragraph does not mention any of the data in this paper and consists only of the cited references. These are the contents that should be written in the introduction. It should be deleted entirely or moved to the introduction.
Author Response
Dear reviewer,
Many thanks for your constructive suggestions of our research, which is very necessary to improve the scientific rigor and readability of the manuscript. We have made serious revisions to our manuscript. Responses to reviewer are as follows:
- Are the NILs used in the experiments truly Near Isogenic Lines?
The authors claim that they constructed two NILs using one molecular marker (L110-113). Alleles of the Wx gene can be identified with only one marker, but it is impossible to distinguish how far the donor parent's chromosomal region extends. If the authors used only one marker for the selection of NILs, there is no guarantee that the NIL will be produced even if nine rounds of BC are performed. Information needs to be given as to how much of the donor parent's chromosomal region these NILs have.
Response: The result of genome resequencing analysis showed that the construction of near isogenic lines was successful. We have added the results of resequencing analysis in the manuscript (line 113-114) and Supplementary Figure 1.
- Were the rice grains properly polished?
In the photograph of Fig. 3A, all grains in HT condition have brown areas. These grains don't seem to be polished enough. Machine for measuring the chalkiness degree of grains generally identifies white immaturity areas based on the light transmitted through the grains. Accurate measurement is impossible when a large amount of rice bran remains in samples. The data on the Chalkiness degree in the HT condition are unreliable.
Response: The long-time of high temperature treatment during the whole filling period in this study resulted in the thickening of aleurone layer in all rice grains. In order to comprehensively present the effect of high temperature on rice grain, grains with partially unpolished grains were exhibited our manuscript. Completely-polished grains (get rid of the yellow aleurone layer) were used for physical and chemical quality analysis but not for photography, because deep polishing resulted in significantly smaller broken grains, which are not suitable for photography. If it may cause misunderstanding of readers, we can consider deleting this part of data.
The important problems of this manuscript are pointed out below.
L104-122
Not enough information on growing conditions for plant materials. Write down the sowing date (or the transplanting date), heading date for each line, amount of applied fertilizer, bucket size, and number of plants grown per bucket. The minimum temperature (or average temperature) in the growth chamber should be indicated in addition to the maximum temperature.
Response: We have added more information about the growing conditions for plant materials (line 120-128).
L198-200 “Under normal temperature, ... from 2661-Wxb.”
The only data available were photographs (Fig. 2A), and I was unable to read the results claimed by the author from these photographs. The authors should present the results objectively by quantifying the length, width, weight, etc. of the growing grains.
Response: Generally, the change of chlorophyll content in seed coat (the change of green shade) can be used to judge the filling process of rice grains, but it’s not applicable be used to compare the filling progresses of different rice varieties, so we deleted the conclusion here (line 204-207).
Fig. 2B
While the 1,000-grain weight of the 3611 background Wxa in NT is about 22 g, the weight in HT appears to be about only 14 g. In general, normally ripened rice grains don’t produce such a large difference in 1,000-grain weight. It can be inferred that the 1,000-grain weight was measured including the rice that had stopped ripening. Thousand grain weight should be measured by fully ripened rice. Sterile grains should be removed and remeasured.
Response: In our study, the high temperature treatment was carried out throughout the whole grain-filling period, which is a relatively intense and long-lasting treatment. Hence, the high temperature treatment had a very obvious influence on the grain weight. The seeds we used for the 1000-grain weight measurement were all fully pollinated and ripened grains
L217-220 “The chalky area of ..., which lead to chalky production.”
This is not the content that should be noted in the results.
Response: We have deleted the part of this content (line 223).
Fig. 6
Data such as BD and PK shown in Supplementary Table 2 are important for the analysis of starch properties using RVA. This figure should be moved to Supplementary data and Supplementary Table 2 should be included in the text. However, the experimental data in Supplementary Table 2 do not appear to have been measured multiple times. Perform replicate experiments and show the mean and the results of the significance test.
Response: The statistical data of RVA eigenvalues were not presented in Supplementary data before due to negligence. We have moved the statistical table of RVA eigenvalues to the main text as Table 1, and transferred the figure of RVA curves to the Supplementary data as Supplementary Figure 4.
L362-393
This paragraph does not mention any of the data in this paper and consists only of the cited references. These are the contents that should be written in the introduction. It should be deleted entirely or moved to the introduction.
Response: We have deleted most of this paragraph, part of which was integrated into other parts of the discussion (line 369).

Round 2
Reviewer 2 Report
Dear authors,
I feel that this revision has greatly improved the manuscript. However, it was not possible to dispel doubts about the reliability of the data. A critical problem of this manuscript is pointed out below.
Original comment:
2. Were the rice grains properly polished?
In the photograph of Fig. 3A, all grains in HT condition have brown areas. These grains don't seem to be polished enough. Machine for measuring the chalkiness degree of grains generally identifies white immaturity areas based on the light transmitted through the grains. Accurate measurement is impossible when a large amount of rice bran remains in samples. The data on the Chalkiness degree in the HT condition are unreliable.
Your response: The long-time of high temperature treatment during the whole filling period in this study resulted in the thickening of aleurone layer in all rice grains. In order to comprehensively present the effect of high temperature on rice grain, grains with partially unpolished grains were exhibited our manuscript. Completely-polished grains (get rid of the yellow aleurone layer) were used for physical and chemical quality analysis but not for photography, because deep polishing resulted in significantly smaller broken grains, which are not suitable for photography. If it may cause misunderstanding of readers, we can consider deleting this part of data.
New comment: Was the chalkiness degree measured with broken rice in this experiment? I have not used the same scanner as the authors, but I have experience with similar one. As far as I know, broken rice will not give an accurate reading. I must conclude that there are major doubts about the reliability of the data. Please show the evidence that accurate measurement is possible even with broken rice. If perfect polishing cannot be performed, the chalkiness degree should be measured in brown rice for both HT and NT.
Other fixes are as follows.
Fig. 3A
If the authors want to keep the picture on the manuscript, please write in the caption that the HT grains were weakly polished than the rice for physical and chemical quality analysis to maintain its shape.
Original comment:
L104-122
Not enough information on growing conditions for plant materials. Write down the sowing date (or the transplanting date), heading date for each line, amount of applied fertilizer, bucket size, and number of plants grown per bucket. The minimum temperature (or average temperature) in the growth chamber should be indicated in addition to the maximum temperature.
Your response: We have added more information about the growing conditions for plant materials (line 120-128).
New comment: Please add information about the heading date of the four lines used in the experiment. The information for heading data is very important. It is possible that different lines have different heading dates, not only between lines with different genetic backgrounds, but even between NILs. If the heading dates were different, please indicate whether the start of high temperature treatment was different for each line or if it started on the same day.
Original comment:
L198-200 “Under normal temperature, ... from 2661-Wxb.”
The only data available were photographs (Fig. 2A), and I was unable to read the results claimed by the author from these photographs. The authors should present the results objectively by quantifying the length, width, weight, etc. of the growing grains.
Your response: Generally, the change of chlorophyll content in seed coat (the change of green shade) can be used to judge the filling process of rice grains, but it’s not applicable be used to compare the filling progresses of different rice varieties, so we deleted the conclusion here (line 204-207).
New comment:
L205-208 “High temperature accelerated ... under normal temperature”
It is true that high temperatures generally accelerate the growth of grains, but this knowledge cannot be read from Fig. 2A. If you want to discuss such a subject, you should write to Discussion using citations.
Furthermore, I don't understand why you mention photosynthesis here. Does the brown rice in the husk carry out photosynthesis?
Original comment:
Fig. 2B
While the 1,000-grain weight of the 3611 background Wxa in NT is about 22 g, the weight in HT appears to be about only 14 g. In general, normally ripened rice grains don’t produce such a large difference in 1,000-grain weight. It can be inferred that the 1,000-grain weight was measured including the rice that had stopped ripening. Thousand grain weight should be measured by fully ripened rice. Sterile grains should be removed and remeasured.
Your response: In our study, the high temperature treatment was carried out throughout the whole grain-filling period, which is a relatively intense and long-lasting treatment. Hence, the high temperature treatment had a very obvious influence on the grain weight. The seeds we used for the 1000-grain weight measurement were all fully pollinated and ripened grains
New comment:
Fully ripened grains generally are defined as spikelet that fully filled, or brown rice derived from such spikelet. If brown rice with a 1,000-grain weight that is 39% lower is fully ripened grains, the volume of the husk should also be reduced by nearly 39%. High temperature treatment started 5 days after flowering, and it is impossible that the volume of husk is different between HT and NT. Therefore, it is reasonable to conclude that the brown rice in HT underwent intense and long high-temperature treatment and stopped growing during enlargement. Please show reasons why the authors believe these grains to be fully ripened grains.
Alternatively, I recommend adding in Materials and Methods and in the figure caption that the grains used for physical and chemical quality analysis have been included grains that were not fully ripe. It is also necessary to pay attention to the properties of the sample grains in this experiment when conducting Discussion.
Author Response
Dear reviewer,
Thanks again for your constructive suggestions of our research, which is very necessary to improve the scientific rigor and readability of the manuscript. We have made serious revisions to our manuscript. Responses to reviewer are as follows:
We did't understand the meaning of “fully ripened grains” before. We wanted to indicate that the high temperature treatment carried out 5 days after flowering could ensure the normal pollination of rice. After removing empty-shell grains, all other flower-marked seeds were collect to measure grain weight, and we cannot guarantee that all grains were ripen fully.
We have noted in the figure caption that grains that not fully ripened were also included for weight measurement (line217-218). In addition, we used your viewpoint in the discussion section to summarize the results of high temperature influence on thousand-kernel weight (line374-380).